# Assessment of the Genotoxic and Cytotoxic Effects of Turpentine in Painters

**DOI:** 10.3390/life13020530

**Published:** 2023-02-15

**Authors:** Sara Kević Dešić, Barbara Viljetić, Jasenka Wagner

**Affiliations:** 1Department of Medical Biology and Genetics, Faculty of Medicine, Josip Juraj Strossmayer University, 31000 Osijek, Croatia; 2Department of Medical Chemistry, Biochemistry and Clinical Chemistry, Faculty of Medicine, Josip Juraj Strossmayer University, 31000 Osijek, Croatia

**Keywords:** turpentine, micronucleus tests, genotoxicity tests, peripheral blood lymphocytes

## Abstract

Turpentine is a fluid used mainly as a solvent for thinning oil-based paints, obtained by distilling the resin of coniferous trees. Fine art painters use turpentine on a daily basis. The aim of this study was to investigate the genotoxic effect of turpentine and to determine the lymphocyte proliferation index in the peripheral blood of individuals occupationally exposed to turpentine. For this purpose, the cytokinesis-block micronucleus assay (CBMN) was used to determine the total number of micronuclei (MNi), nucleoplasmic bridges (NPB), and nuclear buds (NBUD), as well as the cell proliferation index (CBPI) in the peripheral blood lymphocytes of the subjects. Twenty-two subjects exposed to turpentine daily through their work participated in the study and were compared to twenty subjects in the control group. The results showed a significant increase in the number of micronuclei and other genotoxicity parameters, as well as significant cytotoxicity based on CBPI values. In addition, the genotoxic and cytotoxic effects of turpentine were found to be time-dependent, i.e., the deleterious effects of turpentine on genetic material increase with prolonged exposure. These results strongly suggest that exposure to turpentine vapors may affect genome stability and that occupational safety measures should be taken when using turpentine.

## 1. Introduction

Substances that can have a harmful effect on health are very often used in professional work. One such substance is turpentine, which is used by artists in their daily work as a solvent for paints and to whose vapors they are exposed. Turpentine (C_10_H_16_) is a transparent, volatile, colorless liquid with a bitter taste and characteristic odor, obtained by distilling resin from coniferous trees, primarily pine, larch, or cedar. It consists mainly of the monoterpenes alpha-pinene and beta-pinene and smaller amounts of carene, camphene, dipentene, and terpinolene [1]. It is not soluble in water, but dissolves in alcohol, ether, and chloroform [2] and is highly flammable.

Turpentine is used primarily as a solvent for thinning oil-based paints and is therefore widely used by fine art painters, but it is also used in other fields as an organic solvent. It is also used as a raw material for the production of protective varnishes and waxes for wood, and as a raw material for the production of camphor and methanol. In very small quantities, it is also used as a flavoring agent for food and beverages. Its use in medicine has a long tradition. There it is used mixed with animal fat as a healing ointment and is still a component of similar preparations for external use [3]. The antibacterial activity of turpentine against *Staphylococcus epidermidis* and *Escherichia coli* was demonstrated in an in vitro study [4], and the phytomedical significance of turpentine nanoemulsions against methicillin-resistant *Staphylococcus aureus* was demonstrated [5]. Since it has antiparasitic activity, turpentine has been used for the treatment of myiasis [6]. Thanks to its antiseptic and diuretic properties, it has even been used as a remedy against intestinal parasites and candidiasis [7].

In addition, however, numerous adverse effects of turpentine on human health are known, the intensity of which depends on the type of exposure. Exposure to turpentine can cause skin, eye, and respiratory irritation, as well as allergies due to a hypersensitivity reaction type IV. Alphapine, delta-3-carene, and turpentine peroxides are the major allergens of turpentine. There are numerous reports of contact dermatitis caused by exposure to turpentine [8], but there are also reports of pulmonary necrosis due to inhalation [9] or poisoning due to ingestion [10].

However, despite all the listed known adverse effects, it is not known whether turpentine is a mutagen or a carcinogen. Therefore, the cytotoxic and genotoxic effects of turpentine on human peripheral blood lymphocyte cultures in individuals occupationally exposed to turpentine, particularly turpentine vapors, were examined for the first time in this study. For this purpose, the in vitro cytotoxic and genotoxic activities of turpentine were evaluated using micronucleus assays (MN). In addition to the number of micronuclei in binuclear cells, the number of nucleoplasmic bridges, indicating the presence of dicentrics and the number of nuclear buds, representing gene amplification, were also evaluated. To determine cytotoxicity, the cell proliferation index (CBPI) of 500 cells is calculated. The CBPI is a number that indicates the cell division in the subject compared to the control.

## 2. Materials and Methods

The present study was a cohort study approved by the Ethics Committee of the Faculty of Medicine, Josip Juraj Strossmayer University of Osijek (approval number 2158-61-07-16-17 from 8 April 2016). The study was conducted with a group of 42 participants (28 women and 14 men) aged 18–28 years, mostly students.

The group of subjects studied consisted of individuals who were exposed to turpentine vapors on a daily basis, i.e., students of the Department of Fine Arts at the Academy of Arts in Osijek, J. J. Strossmayer University of Osijek. The criteria for inclusion in the group of subjects were that they were under 40 years of age, that they had been exposed to turpentine, and that they had not been exposed to ionizing radiation in the last six months before the study or to antibiotic therapy in the last month. After providing written informed consent, subjects also completed a questionnaire about their smoking habits, alcohol consumption, health status, family history of cancer, past or current medication use, and diagnostic procedures. No private details about subjects participating in the studies were or will be released to the public. Twenty-two subjects who met the specified criteria participated in the study, and a corresponding control group was formed.

The control group consisted of individuals of similar age, sex, and lifestyle as the test group, with the condition that they were not exposed to turpentine. The subjects in the control group also gave informed consent and, as with the test subjects, completed a questionnaire about their lifestyle habits. On this basis, the suitability of the control group was checked. The criteria for inclusion in the control group were that the subjects were under 40 years of age, had not been exposed to turpentine, had not been exposed to ionizing radiation or any other chemical substance for at least six months before participating in the study, and had not taken antibiotics in the past month. Twenty subjects in the control group participated in the study, and their eligibility was subsequently evaluated using statistical methods.

Chemicals and media used in this study: F-10 Ham medium was from EuroClone, Pero, Italy; L-glutamine was from Lonza, Basel, Switzerland; cytochalasin-B, methanol, and glacial acetic acid were from Sigma, St. Louis, MO, USA; phytohemagglutinin and Gurr Buffer Tablets were from Gibco, Paisley, Scotland, United Kingdom; Giemsa dye was from Kemijsko tehnički laboratorij Šlaković, Zagreb, Croatia; heparinized vacutainer tubes were from Becton Dickinson, Franklin Lakes, NJ, USA. All of the other reagents used were laboratory-grade chemicals from Kemika, Zagreb, Croatia.

Peripheral blood was collected by venipuncture from 8 November 2016 to 6 December 2016 in sterile heparinized tubes at the Laboratory of Medical Genetics, Faculty of Medicine Osijek. All blood samples collected were handled in the same way and processed as soon as possible, not more than four hours after blood collection.

The cytokinesis-block micronucleus (CBMN) assay was performed as described by the HUMN project [11], with minor modifications to the protocol. A total of 500 μL of whole blood was added to F-10 Ham medium supplemented with phytohemagglutinin and L-glutamine and incubated for 44 h at 37 °C and 5% CO_2_. After 44 h of incubation, Cytochalasin-B was added to each sample at a final concentration of 6 μg/mL to prevent cytokinesis. After 28 h of Cytochalasin-B, for a total of 72 h of incubation, cells were harvested. Lymphocytes were fixed in a methanol–acetic acid solution, air dried overnight, and stained with 5% Giemsa solution.

Microscopic analysis was performed using a light microscope (Zeiss Axioscope 2 MOT) with a final magnification of 1000×. Each subject was analysed for the total number of MNi, NPBs, and NBUDs per 1000 binucleated cells (BNCs) according to the criteria published by Fenech [12]. Only BNCs with well-preserved cytoplasm were evaluated for analysis. The frequencies of mononuclear, binucleated, and multinucleated cells were also evaluated in 500 cells per subject. CBPI was calculated on the same slides using the formula: (M1 + 2M2 + 3M3 + 4M4)/500, where M1-M4 represent the number of cells with one to four nuclei [13].

## 3. Results

### 3.1. Characteristics of the Population and Review of the Control Group

This study included a group of 42 participants (28 women and 14 men) aged 18–28 years (median age 22 years). Detailed population and lifestyle characteristics are shown in Table 1. The group of subjects exposed to turpentine were students of the Academy of Arts and Culture in Osijek, who use turpentine as a solvent for painting in their daily work. None of the subjects studied or the control group had been exposed to ionizing radiation for at least six months or had received antibiotic therapy for at least one month before blood sampling.

To measure genome damage in subjects occupationally exposed to turpentine, we had to ensure that we had a suitable control group and that the control group was similar to the tested group in all factors that might affect the results except exposure to turpentine. Twenty subjects in the control group participated in the study, and their adequacy was evaluated by statistical methods. Fisher’s exact test was used to compare the control group and the exposed group to check the accuracy of the control group. It showed that there was no statistically significant difference in gender, smoking and alcohol consumption habits, and family history of cancer, which are important factors for micronuclei formation. The Mann–Whitney U test was used to compare the age of the control group with that of the exposed group, and it also showed no statistically significant difference (Table 2).

### 3.2. Influence of Occupational Exposure on the Parameters of the CBMN Test

The CBMN assay was used to assess the frequency of MNi, NPBs, and NBUDs associated with cell proliferation as measured by CBPI. The results of our cytogenetic biomonitoring study, divided into two categories: the turpentine-exposed group and the control group, are shown in Figure 1. The median MNi frequency for the control group was 0 with IQR 0–1.25 per 1000 BNCs and ranged from 0 to 5 (11 out of 20 control samples had 0 MNi detected), and for the turpentine-exposed group, the median MNi frequency was 7.50 with IQR 5.25–12.5 per 1000 BNCs and ranged from 0 to 22 (where only one sample had 0 MNi). The median frequency of NPBs in the control group was 0 per 1000 BNCs and ranged from 0 to 2, whereas the median frequency of NPBs in the exposed group was 0 with IQR 0–1 per 1000 BNCs and ranged from 0–3. The median frequency of NBUDs in the controls was 0 with IQR 0–0.25 per 1000 BNCs and ranged from 0 to 2, while the median frequency of NBUDs in the exposed group was 1 with IQR 0–5.5 per 1000 BNCs and ranged from 0–9. As for cell proliferation, the median CBPI was 2.28 with IQR 2.2–2.38 and ranged from 2.05 to 2.47 for controls and 2.73 with IQR 2.52–2.86 and ranged from 2.25 to 3.11 for turpentine exposure.

Comparison of the data for MNi, NPBs, and NBUDs between the control population and the turpentine-exposed students showed a significant difference in all measured parameters of genotoxicity and a statistically significant difference in cytotoxicity according to CBPI (*p* < 0.05) based on the Mann–Whitney U test.

### 3.3. Influence of Duration of Turpentine Exposure and Combination of Turpentine with Other Substances on Parameters of CBMN Test

To evaluate the influence of the duration of exposure to turpentine on the genetic damage of micronuclei, the exposed group was divided into three subgroups: exposure period 1 includes subjects who were exposed for the shortest time (less than one year; n = 4), exposure period 2 includes subjects who were moderately exposed (one to two years; n = 6), and exposure period 3 includes subjects who were exposed to turpentine for the longest time (2 years or longer; n = 12). The median values and IQR of CBMN assay parameters for each subgroup are summarized in Table 3. The analysis showed statistically significant differences in almost all parameters analyzed (except for the NPBs parameter) as a function of exposure time.

Because some subjects within the tested group were also exposed to other chemicals in their work that could potentially affect the result, such as nitric acid and nitro thinner, we also tested whether there was a statistically significant difference in genotoxicity and cytotoxicity between subjects exposed only to turpentine (n = 6) and those who also used other agents (n = 16). The median values with IQR and distributions of the parameters of the CBMN test for the subjects exposed to turpentine only and the subjects who also used other agents are summarized in Table 4. The analysis showed no statistically significant differences in the CBMN parameters between the subjects who were exposed not only to turpentine but also to other chemicals.

External factors such as smoking habits and alcohol consumption that might affect the number of parameters of the CBMN test were also considered. The median values with IQR and baseline statistics for total MNi, NPB, NBUD, and CBPI for the selected subjects according to their lifestyle are shown in Table 4. Due to the correlation of smoking and alcohol consumption, there is no statistically significant difference for any of the tested parameters of the CBMN test (total number of MNi, NPB, NBUD, and CBPI) in subjects with alcohol (n = 17) and smoking habits (n = 9).

## 4. Discussion

Assessment of the genotoxic effects of chemicals is performed on human peripheral blood lymphocytes using various genetic markers that allow early detection of biological effects. One of these genetic markers is the number of micronuclei in binuclear cells [11] and the change in CBPI, which is used as an indicator of a cytotoxic effect. The simplicity and sensitivity of this cytogenetic technique makes it the method of choice in studies assessing genetic damage, in this case to investigate the genotoxic effect of turpentine.

Turpentine is a substance very commonly used as a solvent, so the aim of this study was to verify the genotoxic effect of turpentine vapor by counting micronuclei in the lymphocytes of painters. Since the micronuclei values obtained for the general Croatian population [14] are not suitable as a reference point for this study because the subjects were significantly older compared to the group used in this study, a statistical evaluation of the adequacy of the control group was performed. Age, sex, smoking habits, and family history of cancer, known to affect the number of MN, were tested [15,16,17], and it was found that there was no statistically significant difference in the above parameters, i.e., the adequacy of the control group was confirmed.

It has long been known that fine art painters often suffer from various diseases (skin and lung diseases, tendency to mental disorders) [8,18,19] related to the use of various solvents, pigments, media, fixatives, and even the paints themselves, which until recently contained high levels of heavy metals, especially lead, which has been shown to be extremely genotoxic [20]. Because turpentine remains one of the most commonly used solvents in the daily work of fine artists, this study examined the toxic effects of turpentine on the human genome and showed that turpentine causes the formation of a greater number of micronuclei in binuclear cells in individuals exposed to turpentine vapors on a daily basis. In addition to the increased number of MN, other parameters indicative of a genotoxic effect were also observed, and a greater number of nuclear buds as a result of gene amplification and a greater number of nuclear bridges as a result of dicentric formation were detected [21]. There was also a statistically significant difference in the evaluation of the cytotoxic effect of turpentine by calculating the CBPI, which indicates the antiproliferative effect of turpentine (Table 3).

The main constituent of turpentine, alpha-pinene, is a naturally occurring bicyclic and hydrophobic monoterpene commonly used in insecticides, solvents, perfumes, and as a food additive. However, despite its widespread use, there are few data on the safety assessment of alpha-pinene [22,23]. Therefore, Catanzaro et al. investigated the cytotoxic and genotoxic effects induced by acute exposure of alpha-pinene to Chinese hamster V79-Cl3 cells [24]. They showed that exposure to increasing concentrations of alpha-pinene significantly increased the frequency of micronucleated and multinucleated cells and that DNA damage was most likely due to the increased production of reactive oxygen species (ROS). Although the effect of pure alpha-pinene has been demonstrated, it could be assumed that the genotoxic effect of turpentine is caused by alpha-pinene since it is the main component of turpentine used as a solvent. In contrast, alpha-pinene was not found to be mutagenic in vitro or in vivo in studies conducted by the US National Toxicology Program (NTP) [25]. The study was conducted in rats and mice under conditions of three months of inhalation of alpha-pinene up to 400 ppm. Neither male nor female mice exposed to α-pinene showed an increase in the percentage of polychromatic erythrocytes as a biomarker of chromosomal damage in peripheral blood, indicating that there is no bone marrow toxicity. This suggests that alpha-pinene was either not toxic to bone marrow or did not reach the bone marrow compartment. In the same study, it was also found that alpha-pinene was negative for mutagenicity in the bacterial gene mutation assay and with and without induced rat S9 fraction in *Escherichia coli* and *Salmonella typhimurium* [25]. However, monoterpenes structurally related to alpha-pinene are carcinogenic, and because there is a lack of human toxicity data to characterize the potential hazard of exposure to α-pinene, the NTP is investigating the chronic toxicity and carcinogenicity of α-pinene following whole-body inhalation exposure of rats (up to 200 ppm) and mice (up to 400 ppm) to alpha-pinene [26]. In a recent study, the toxicokinetic evaluation of alpha-pinene as a common indoor air pollutant oxide was performed after inhalation exposure in rodents. With sex-dependent toxicokinetic behavior and high tissue retention, the formation of alpha-pinene oxide from the metabolism of alpha-pinene in vivo was reported [27]. The results indicated that the formation of alpha-pinene oxide was dose-dependent, with a higher ratio observed at low dose than at high dose, suggesting a possible saturation of metabolism with an increasing dose. Subsequent analysis of alpha-pinene and its potentially reactive metabolite alpha-pinene confirmed that alpha-pinene was not mutagenic in *S. typhimurium* or *E. coli* with or without induced rat liver S9, whereas alpha-pinene oxide was mutagenic [28]. This result is very interesting and, as the authors noted, could possibly explain the lack of mutagenicity of alpha-pinene.

Apart from the arts, paints, which are often complex mixtures that may themselves contain toxic components, are also used in industry in combination with other toxic substances, especially solvents. For example, a study conducted in southern India examined the number of micronuclei in workers in the textile industry who, as an isolated population, were occupationally exposed to various toxic dyes, bleaches, salts, acids, alkalis, and heavy metals, as well as various organic solvents and fixatives used to dye fabrics [29]. Twenty-five subjects of different age, gender, lifestyle, and seniority in the textile industry participated in the above study and a statistically significant difference was found in the number of micronuclei compared to the control group, which is another indicator of the potentially harmful effect of similar substances on the human genome.

In our study, temporal exposure to turpentine was also observed. Participants were divided into three groups according to the duration of exposure—the first group includes individuals who have been working with turpentine professionally for less than one year, the second group includes individuals who have been working with turpentine for more than one but less than two years, while the third group includes individuals who have been working with turpentine daily for more than two years. Analysis of the exposure to turpentine over time showed that with longer and more intense exposure, all parameters of genotoxicity and cytotoxicity increase. Formaldehyde, for example, is a chemical widely used in industry. Several studies have shown that formaldehyde affects the human genome using the micronucleus test, i.e., the population occupationally exposed to formaldehyde has a statistically significant increased number of MN in up to 62% of cases [30]. It should be noted that formaldehyde is classified as a first-in-class human carcinogen by the World Health Organization and that an increase in micronucleus test results has been observed with prolonged exposure, indicating a potentially cumulative effect on genome stability with chronic exposure [31]. A similar result was observed in our study, which may also indicate a cumulative genome stability effect with chronic exposure to turpentine.

The damage of genetic material detected by the micronucleus test in individuals occupationally exposed to turpentine can be largely attributed to the effect of the turpentine itself, as no statistically significant difference in the number of MNi, NPBs, NBUDs, and CBPI was detected between individuals who used turpentine alone and individuals who used both turpentine and other potentially toxic substances (nitric acid and nitro thinner), as shown in Table 4. Research has shown that daily use and occupational exposure to various substances, such as gasoline, heavy metals, and coal dust [32,33] or anesthetics in medicine [34,35], should be carried out with caution and that there is a need to increase awareness of the importance of safety in the workplace. The harmful effects of a chemical need not be immediately apparent but can still have long-term and significant effects on human health, taking into account not only occupational exposure but also everyday exposure to genotoxic substances in our environment, such as the effects of secondhand smoke and smog [36,37] or the use of pharmaceuticals [38,39,40]. For this reason, it was tested whether the habits of smoking and alcohol consumption have an effect on the number of micronuclei, but no association was found. The habit of smoking alone does not increase the number of micronuclei, except in heavy smokers who are not occupationally exposed to genotoxic substances [15].

In summary, this research has proven that daily exposure to turpentine has a genotoxic effect and leads to an increase in all parameters indicative of genotoxicity: the number of micronuclei, nucleoplasmic bridges, and nuclear buds. Turpentine may therefore have a mutagenic and potentially carcinogenic effect, as chromosomal aberrations determined by the micronucleus test have been shown to precede dysplasia and malignant changes [11,41]. On the other hand, it should be kept in mind that, despite the clear results, this study was performed on a relatively small number of subjects and it would be appropriate to continue the study with a larger number of subjects in order to validate the presented results.

## Figures and Tables

**Figure 1 life-13-00530-f001:**
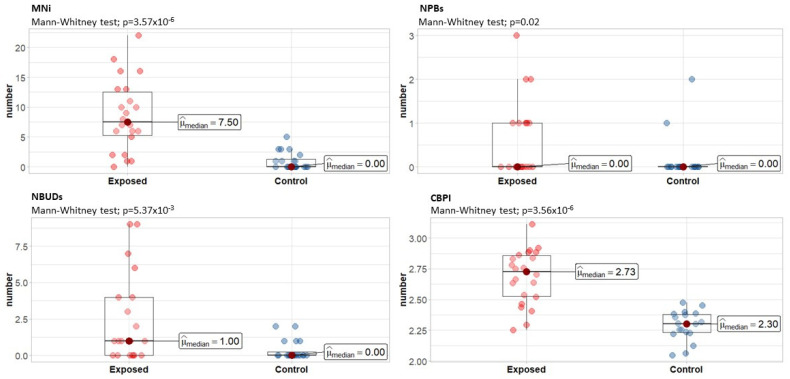
Cytokinesis-block micronucleus assay parameters and cell proliferation index in peripheral lymphocytes of the turpentine-exposed group compared with the control group per 1000 cells. Statistically significant difference (*p* < 0.05) using Mann–Whitney U test was detected in all compared data sets. MNi—micronuclei; NPBs—nucleoplasmic bridges; NBUDs—nuclear buds; CBPI—cytokinesis-block proliferation index.

**Table 1 life-13-00530-t001:** General characteristics of subjects enrolled in the study (median and IQR for the years).

	Women	Men	Total
n	28	14	42
Age (years)	22.5 (20.75–23)	22 (19.5–26)	22 (20.75–23.75)
Age (range years)	19–28	18–28	18–28
Smokers	12	4	16
Alcohol drinkers	21	12	33
Medication exposure	2	3	5
Radiation exposure	2	3	5
Cancer history	9	11	20

**Table 2 life-13-00530-t002:** Adequacy of control group (median and IQR for the years).

Variable	Controls	Turpentine-Exposed	*p*-Value
Sex (M/F)	9/11	5/17	0.19 *
Alcohol drinkers (Yes/No)	16/4	17/5	>0.99 *
Smokers (Yes/No)	8/12	8/14	>0.99 *
Cancer history (Yes/No)	12/8	9/13	0.35 *
Age (Range)	22.5 (21–24)	24 (23–25)	0.11 **

* Fisher exact test and ** Mann–Whitney U test.

**Table 3 life-13-00530-t003:** Median and IQR and basic statistical parameters calculated for the cytokinesis-block micronucleus (CBMN) assay parameters (total number of micronuclei, nucleoplasmic bridges, and nuclear buds per 1000 cells and CBPI) in peripheral blood lymphocytes of the subjects with different durations of turpentine exposure.

	Exposure Time 1	Range	Exposure Time 2	Range	Exposure Time 3	Range
MNi	1.50 (0.75–3.5)	0–2	6.50 (5.25–11.5)	2–22	10 (6.75–13.75) *	0–18
NPBs	0 (0–0.5)	0–2	0 (0–0.75)	0–1	1 (0–1.25)	0–8
NBUDs	0 (0–0.25)	0–1	1.50 (1–5)	0–7	3.5 (0–9) *	0–16
CBPI	2.38 (2.28–2.48)	2.29–2.52	2.65 (2.56–2.86)	2.25–3.11	2.81 (2.74–2.87) *	2.44–2.92

MNi, micronuclei; NPBs, nucleoplasmic bridges; NBUDs, nuclear buds; CBPI, cytokinesis-block proliferation index. * Statistically significant difference *p* < 0.05. Mann–Whitney U test.

**Table 4 life-13-00530-t004:** Median and IQR and basic statistical parameters calculated for the cytokinesis-block micronucleus (CBMN) assay parameters (total number of micronuclei, nucleoplasmic bridges, and nuclear buds per 1000 cells and CBPI) in peripheral blood lymphocytes of the subjects exposed to only turpentine and turpentine in combination with other substances (chemicals, alcohol and smoking).

	Turpentine Only	Range	Turpentine and Other Chemicals	Range	Turpentine and Alcohol	Range	Turpentine and Smoking	Range
MNi	7.5 (2–10)	0–22	7.5 (6–13)	1–22	7 (5–11)	0–18	9 (6–10)	1–18
NPBs	0.5 (0–1)	0–8	0 (0–1)	0–8	1 (0–1)	0–8	1 (0–2)	0–8
NBUDs	1 (0.25–1.75)	0–16	2 (0–6.25)	0–16	1 (0–4)	0–16	1 (0–4)	0–12
CBPI	2.64 (2.55–2.73)	2.25–3.11	2.79 (2.52–2.88)	2.25–3.11	2.75 (2.63–2.84)	2.44–2.92	2.76 (2.63–2.84)	2.52–2.92

MNi, micronuclei; NPBs, nucleoplasmic bridges; NBUDs, nuclear buds; CBPI, cytokinesis-block proliferation index.

## Data Availability

The data presented in this study are available on request from the corresponding author. The data are not publicly available due to ethical restrictions.

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
