# Peer review of "Assessment of the Genotoxic and Cytotoxic Effects of Turpentine in Painters"

_life, 2023, doi:10.3390/life13020530_

Round 1
Reviewer 1 Report
The manuscript attempts to evaluate differences in chromosme damage levels between populations exposed to turpentine and control populations in (predominantlty) university students.
Methods and overall study design is good however there are several clear omissions that make the conclusions difficult to support namely:
1) Table 2 Sex differences in control and exposed groups - in the control group there were 9 males and 11 females whereas the exposed group has 5 males and 17 females, which is a significant difference despie the statistical analysis being non significant. Thsi is important when evaluating micronuclei as female donors tend to have a higher background level of MN due to X-chromosome rejection, many commercial testing labs separate male and female donors on studies for this reason. It is probably not enough to explain the large difference seen but highlights that the control and exposed groups were not comparable.
2) Table 3 - MN data, it is difficult to see the spread of data with only means presented, in the case of MN ranges for both control and exposed groups start at zero indicating that there were some non responders in the exposed group, better visualisation of teh data would allow outliers to be identified, in small sample groups such as these, a single high responder could significantly skew the data.
3) table 3 - CBPI, stats show a difference between exposed and control groups for CBPI yet the ranges overlap, see also point 2) for outliers skewing data.
4) Table 3 - general - the SD values are often larger than the means, particulary for the NBUD values, this should be commented on throughout.
5) Other data in the literature - The authors state that there is litte published data on turpentine mutagenicity howver one of the major components (alpha-pinene) was negative when tested for micronuclei in the bone marrow of male and female treated mice (NTP) treated via inhalation up to 400ppm for 3 months. Further study data is mentioned in (https://www.industrialchemicals.gov.au/sites/default/files/Turpentine_Human%20health%20tier%20II%20assessment.pdf) where negative MN studies in lymphocytes and mammalian cel mutation is discussed. Data may be available from these studies.
In its current sate the data do not fuly suppot the conclusions, there are differences in the two groups compared that may have skewed results. The only other study referenced is one which shows positive results in p53 deficient rodent cell lines, the other (negative) data is not mentioned. Whilst an effect is apparent, without seeing the spread of data and potential outliers it is impossible to verify the real differences.
Reviewer 2 Report
This study has shown for the first time that turpentine is potentially genotoxic in humans based on the results of the cytokinesis block micronucleus assay done on peripheral blood lymphocytes of human subjects exposed to turpentine. The manuscript is well-written except that there are a few errors in the text that need to be fixed. In lines 48 and 50 and other places, the names of the bacterial species should be italicized. In line 48, E. coli was spelled wrong. Escheria should be changed to Escherichia.
Reviewer 3 Report
The investigators report on the potential genotoxic effects of turpentine in occupationally exposed humans. The study was performed on artists exposed to turpentine in the course of their work/study and examined markers of genomic damage in peripheral blood lymphocytes. Overall the manuscript was very well written and organized. There are a handful of minor issues specified below, but once those are addressed, the unit should be ready for publication.
Minor Issues:
Pg 1, ln 17 – Use of the term “painters” may carry a different interpretation depending on who is reading the paper. On first instance I envisioned the study being conducted on commercial structural painters or industrial painters. It was only once I got into the introduction that I understood that it was fine artists in an academic training setting that were the subject of the study. It’s not a critical point, but the authors may choose to clarify this point sooner, e.g. in the abstract, to avoid this type of confusion.
Pg 3, ln 108 – I realize that there isn't a perfectly matched scenario for this, but why was a positive control for the CBMN assay not included? This assay can be used to investigate the genotoxicity of agents acutely applied to the test system, so it’s not a difficult process to demonstrate the validity of the assay/proficiency of the investigators in conducting this method.
Pg 4, ln 161 – I assume that the formatting associated with this line is supposed to be similar to the other Table headers.
Pg 4, ln 167 – It was not clarified/mentioned as to what test was used to make the statistical comparisons between exposed and control groups. The only reference is the phrase “Statistically significant difference p<0.05” used in several places. Please note what test was used for these comparisons, as well as those in Tables 4 and 5.
Pg 7, ln 301 – The authors do note here that the number of subjects included in this study was ‘relatively small’. I would like to know whether any power analyses we performed to justify the numbers that they were able to achieve. Certainly there appear to be some robust elevations that are associated with the turpentine exposed group, but the investigators go even further to stratify the subjects across additional groupings such as length of exposure or exposure to other chemical agents. Thus, the numbers get even lower. Is there evidence that this further subdivision is appropriate/valid? At a minimum I would ask the authors to note how many subjects were included in each of these smaller groups described in Tables 4 and 5 so the reader can decide the potential impact of the smaller group sizes.
